# Deterministic and highly indistinguishable single photons in the telecom C-band

Nico Hauser [1] ✉, Matthias Bayerbach[1], Jochen Kaupp[2], Yorick Reum[2], Giora Peniakov [2], Johannes Michl [2], Martin Kamp[2], Tobias Huber-Loyola [2], Andreas T. Pfenning[2], Sven Höfling[2] & Stefanie Barz [1]

Quantum dots are promising candidates for deterministic single-photon sources, yet achieving high photon indistinguishability at telecom wavelengths remains a critical challenge. Here, we report a quantum dot-based single-photon source operating in the telecommunications C-band that achieves a raw two-photon interference visibility of up to $(91.7 \pm 0.2)\%$, thus setting a new benchmark for indistinguishability in this spectral range. The device consists of an indium arsenide (InAs) quantum dot embedded within indium aluminum gallium arsenide (InAlGaAs) and integrated into a circular Bragg grating resonator. We explore multiple optical excitation schemes to optimize coherence and source performance. The demonstration of two-photon interference visibilities exceeding 90% from a quantum-dot emitter in the telecommunications C-band pushes solid-state single-photon sources further towards practical quantum communication and quantum networks.

Generating indistinguishable single photons is a crucial prerequisite for photonic quantum networking and quantum computation[1–4]. For these applications, photon sources must combine high brightness with exceptional photon quality. Key performance metrics include the purity of single-photon states and the indistinguishability of independently generated photons, typically quantified experimentally through two-photon interference. From a technical perspective, the operation wavelength of the source is also of high importance. In particular, for compatibility with existing fiber-optic networks and silicon-based integrated photonic platforms, emission in the telecommunication C-band around 1550 nm is essential[5].

To date, many applications relying on single-photon sources at 1550 nm use spontaneous parametric down-conversion (SPDC)[6–10]. Whilst SPDC sources offer high brightness and excellent photon properties, their photon generation process is inherently probabilistic[11–13]. This fundamental limitation poses a significant challenge for scalability in protocols that require large numbers of photons, such as those used in photonic quantum computing or quantum networking.

In this context, on-demand sources of indistinguishable photons can offer a significant advantage[14,15]. Among the most promising candidates are semiconductor quantum dots (QDs), which have enabled demonstrations of photonic quantum computing, quantum networking, and integrated quantum photonic platforms[16–20]. Most of these advances have been realized using QDs emitting in the 780 nm to 960 nm wavelength range, including experiments involving up to 40 consecutive photons[21]. Such progress has been driven by the availability of sources in this regime that produce highly indistinguishable photons with low multi-photon emission probabilities[22–27].

In the telecommunications C-band, various single-photon sources based on quantum dots have been demonstrated[28–32]. Recently, significant advances have been made by optimizing material composition and growth techniques, together with the exploration of different optical excitation schemes[33,34]. For indium arsenide (InAs) quantum dots embedded in circular Bragg grating (CBG) resonators, raw two-photon interference visibilities of up to 71.9% have been reported[35]. Other approaches, such as quantum dots integrated into planar samples, mesa structures, or tapered nanobeam waveguides, have achieved visibilities up to 72% under continuous-wave excitation[36–38].

[1]Institute for Functional Matter and Quantum Technologies and Center for Integrated Quantum Science and Technology (IQST), University of Stuttgart, Stuttgart, Germany. [2]Julius-Maximilians-Universität Würzburg, Physikalisches Institut, Lehrstuhl für Technische Physik, Würzburg, Germany. ✉e-mail: nico.hauser@fmq.uni-stuttgart.de

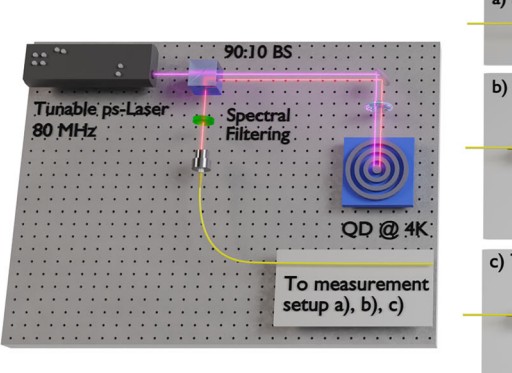
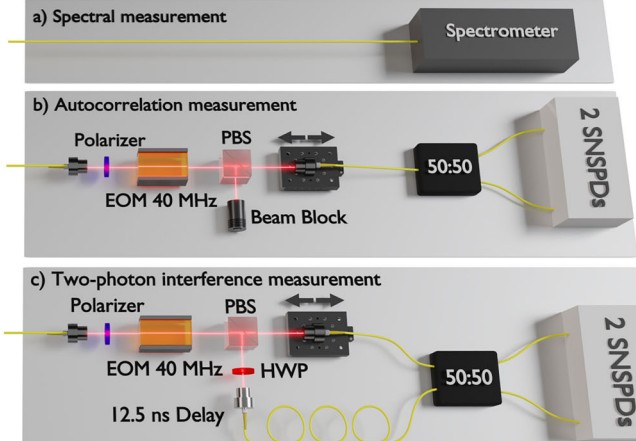

**Fig. 1 | Experimental setup for characterizing the quantum dot emission.** An InAs/InAlGaAs quantum dot (QD) in a circular bragg grating resonator (CBG) inside a cryostat at 4 K is used for photon generation. The QD is excited optically using a ps-laser that is tunable in wavelength and linewidth (repetition rate $\tau_{rep}$ = 80 MHz). The laser is sent to the QD through a 90:10 beam splitter (BS). The QD emission can be spectrally filtered (see Supplementary Fig. 4 for a more detailed sketch of the setup). In order to assess the properties of the photons emitted from the QD, we perform a series of measurements. **a** The generated photons are sent to a spectrometer for spectral characterization. **b**, **c** The generated photons pass an active demultiplexing setup, where consecutively emitted photons can be deterministically separated into two spatial modes using a 40 MHz electro-optical modulator (EOM) and a polarizing beam splitter (PBS). For the autocorrelation measurement (**b**), the photons are then sent to one input of a fiber-based 50:50 beam splitter (whilst the second input is blocked). The output statistics are then measured using time taggers connected to superconducting nanowire single-photon detectors (SNSPDs). For the measurement of two-photon interference (**c**), consecutively emitted photons are sent to either input of the BS and coincidences are recorded.

However, these latter schemes do not qualify as true on-demand sources as there is no temporal information indicating when the quantum dot is excited.

Here, we present a deterministic quantum dot-based single-photon source in the telecommunications C-band demonstrating two-photon interference visibilities exceeding 90%. This represents a key milestone, as it brings QD-based sources into a regime suitable for applications in quantum computing and quantum networking. We systematically investigate different excitation schemes and identify optimal parameters that enable optimal performance. In particular, we achieve a raw two-photon interference visibility as high as $(91.7 \pm 0.2)$% using an incoherent phonon-assisted excitation scheme. This level of indistinguishability, now comparable with probabilistic sources such as SPDC, establishes a new state-of-the-art for deterministic emitters in the telecommunications C-band. Our results mark a crucial step toward scalable photonic quantum technologies based on quantum dots, combining on-demand operation with optimal photon quality.

## Results

### Experiment

Our device uses an InAs/InAlGaAs QD integrated into a circular Bragg grating resonator. The InAs quantum dots were grown by means of gas-source molecular beam epitaxy and fabricated by means of electron beam lithography and dry-chemical etching. Emphasis was put on a refined optimization of the crystal growth by employing ternary digital alloying of the quaternary cladding material, and rotation stop growth calibration. The optimized growth provides a reduced dephasing time, whereas integration into the CBG resonator facilitates reduced excitionic lifetimes[29,33–35,39,40] (see ref. 35 for detailed information on the growth, design and nanofabrication process).

We perform a comprehensive series of characterization measurements, systematically validating the performance of our quantum dot-based source and confirming its suitability for a range of quantum applications.

First, we probe the QD sample with different pump wavelengths to identify relevant resonant emission lines. A pulsed, tunable laser is sent to the QD sample, the generated photons pass a 90:10 beam splitter (BS) and are spectrally filtered to remove any residual pump light. The spectral properties of the photons are analyzed using a spectrometer, tuning the wavelength of the pump laser allows identifying relevant resonances (see Fig. 1a and photoluminescence spectrum in Supplementary Fig. 1).

We then analyze the quality of the generated photons for the different excitation schemes in terms of the photon statistics and the indistinguishability of the generated photons, both crucial properties for any quantum application. To probe the photon statistics of the quantum dot emission, we select a single emission line using a narrow bandpass filter. The second-order autocorrelation is then measured by directing the generated photons to one input of a fiber-based 50:50 beam splitter, detecting coincidences at the two outputs while varying the time delay between the detectors (see Fig. 1b)[41].

To analyze the indistinguishability, we conduct two-photon interference experiments. Consecutively emitted photons are separated using an electro-optic modulator (EOM) and a polarizing beam splitter (PBS)[42]. The PBS outputs are then connected to the two inputs of a fiber-based 50:50 beam splitter, with one output delayed to ensure that the consecutive photons arrive at the splitter simultaneously. A half-wave plate (HWP) in one input allows us to adjust the polarization of the photons, enabling them to have either orthogonal or parallel polarizations. The indistinguishability of the photons is then determined by measuring the coincidence rates at the outputs of the beam splitter for both orthogonal and parallel input polarizations (see Fig. 1c and Supplementary Fig. 4).

### Sample characterization and excitation schemes

The first excitation scheme we explore involves pumping above the band gap with a laser at $\lambda_{pump}$ = 800.0 nm. The resulting photoluminescence spectrum of the quantum dot (see Fig. 2a) shows a dominant line at $\lambda_e$ = 1544.5 nm. This particular emission corresponds to a charged exciton transition (as identified in ref. 35), and we selectively filter this line using a variable bandpass filter in the subsequent experiments (indicated by the green bar in Fig. 2a).

The second excitation scheme we explore involves excitation mediated by longitudinal acoustic (LA) phonons (see Fig. 2b). In this case, the pump laser is set to a wavelength of $\lambda_{pump}$ = 1540.7 nm, slightly blue-detuned from the s-shell resonance at $\lambda_e$ = 1544.5 nm,

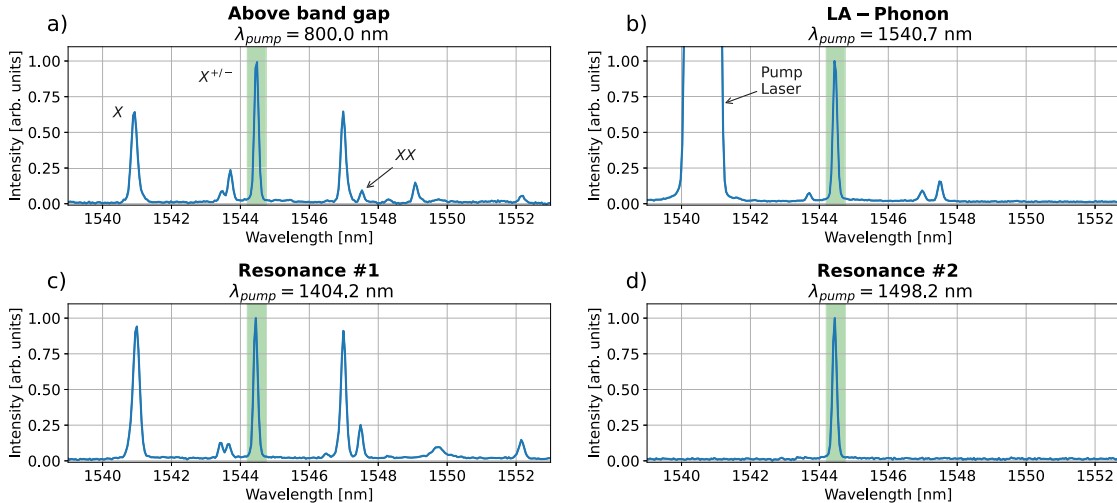

**Fig. 2 | Photoluminescence spectra for different excitation wavelengths.** QD emission spectra shown for (**a**) above-band-gap excitation, (**b**) LA-phonon-assisted excitation, (**c**) resonance #1 ($\lambda_{\text{pump}} = 1404.2$ nm) and (**d**) resonance #2 ($\lambda_{\text{pump}} = 1498.2$ nm). Here, $X$ denotes the neutral exciton, $X^{+/-}$ the charged exciton and $XX$ the biexciton transition (see ref. 35). For the following experiments, the dominant line at $\lambda_e = 1544.5$ nm is filtered using a bandpass filter (green bar).

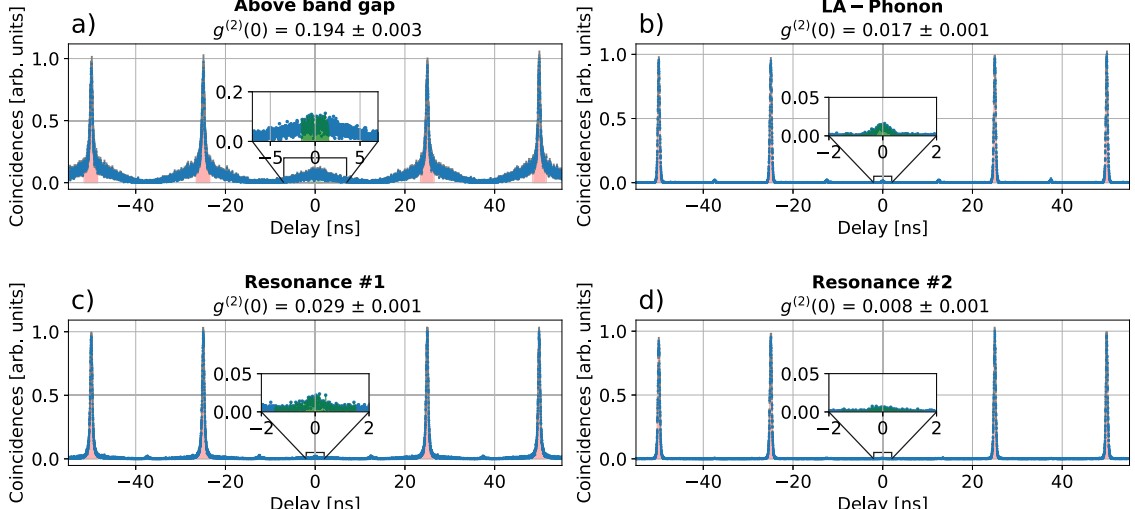

**Fig. 3 | Measurement of photon statistics for different excitation schemes.** We determine the second-order autocorrelation function by measuring coincidences at the output of a 50:50 beam splitter for excitation using (**a**) above-band-gap, (**b**) LA-phonon-assisted, (**c**) resonance #1, and (**d**) resonance #2. It is evident that the correlation peaks measured for above-band-gap excitation as well as at resonance #1 are broadened compared to resonance #2 and LA-phonon-assisted excitation, indicating an increased excited-state lifetime for these resonances. The values $g^{(2)}(\tau = 0)$ are determined by calculating the average area beneath the correlation peaks at $\tau \neq 0$ (highlighted in red) and comparing it to the area around $\tau = 0$ (highlighted in green) in a window of $\Delta t = 3$ ns as shown in the insets. All excitation schemes show $g^{(2)}(\tau = 0) < 0.5$, thus indicating mainly single-photon emission. Note that a peak is present every 25 ns as a result of the active demultiplexing.

enabling excitation of the quantum dot via a phonon sideband. This approach has been demonstrated to be resilient to fluctuations of the pump power while still allowing for a high excited state occupation of the quantum system[43–46].

To identify additional excitation schemes, we perform a wavelength sweep of the pump laser. The resulting spectra are shown in Supplementary Fig. 1, where two strong resonances are observed at $\lambda_{\text{pump}} = 1404.2$ nm (resonance #1) and $\lambda_{\text{pump}} = 1498.2$ nm (resonance #2). The spectra corresponding to all identified resonances are presented in Fig. 2. Importantly, the detuning from the quantum dot emission in each excitation scheme enables effective spectral filtering, thereby minimizing residual pump light.

## Photon statistics

We then proceed to characterize the properties of the generated photons for the different excitation schemes. In order to investigate the photon statistics from the QD emission, the second-order autocorrelation at zero time-delay $g^{(2)}(\tau = 0)$ is determined, which allows us to conclude whether the QD emits single photons or a higher photon number. The measurement is performed by exciting the QD using the different excitation schemes and measuring coincidences at the two outputs of the fiber BS (see Figs. 1b, 3).

To compute $g^{(2)}(\tau = 0)$, we determine two areas using the curves displayed in Fig. 3: the average area $A_{\text{avg}}$ underneath the four coincidence peaks at delay $\tau \neq 0$ ns (highlighted in red in Fig. 3), and the

**Table 1 | Summary of excitation parameters and results for $g^{(2)}(0)$ as well as photon indistinguishability**

| Excitation | $\lambda_{pump}$ | $\Delta E$ | $\Delta\lambda$ | $\Delta\tau$ | $P_{Laser}$ | $g^{(2)}(\tau = 0)$ | $V_{TPI}$ | $M_s$ |
|---|---|---|---|---|---|---|---|---|
| Above band gap | 800.0 nm | 747 meV | 2 nm | 2 ps | 300 nW | 0.194 ± 0.003 | 0.206 ± 0.006 | 0.496 ± 0.013 |
| LA-phonon | 1540.7 nm | 2 meV | 1.0 nm | 6 ps | 170 nW | 0.017 ± 0.001 | 0.917 ± 0.002 | 0.950 ± 0.004 |
| Resonance #1 | 1404.2 nm | 80 meV | 0.8 nm | 8 ps | 70 nW | 0.029 ± 0.001 | 0.581 ± 0.003 | 0.628 ± 0.005 |
| Resonance #2 | 1498.2 nm | 25 meV | 0.5 nm | 13 ps | 22 μW | 0.008 ± 0.001 | 0.845 ± 0.005 | 0.860 ± 0.006 |

The table shows the pump laser parameters (central wavelength $\lambda_{pump}$, detuning from emission line $\Delta E$, linewidth $\Delta\lambda$, pulse duration $\Delta\tau$ and power $P_{Laser}$) used for the quantum dot excitation, which were set using a tunable pump laser along with a pulse slicer. The correlation measurements for obtaining $g^{(2)}(\tau = 0)$, $V_{TPI}$ and $M_s$ have been performed at $P_{Laser} \approx 0.1\ P_{Saturation}$.

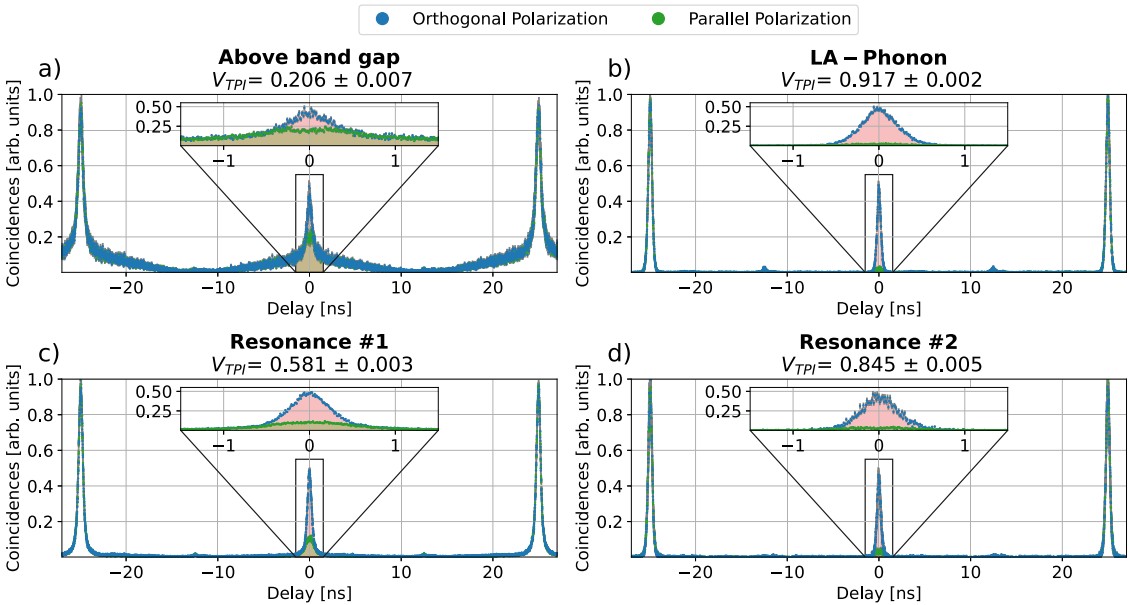

**Fig. 4 | Two-photon interference measurements.** Coincidences measured for photons incident on a 50:50 beam splitter with parallel (green) and orthogonal (blue) polarizations for (**a**) Above-band-gap excitation, (**b**) LA-phonon-assisted excitation, (**c**) excitation using resonance #1 and (**d**) excitation using resonance #2. The coincidence counts are normalized using the area of the coincidence peaks at area $A_0$ at $\tau = 0$ (highlighted in green in Fig. 3). The areas determined within a window of $\Delta t = 3$ ns to account for the whole width of the peaks. The second-order autocorrelation at zero time delay $g^{(2)}(\tau = 0)$ is then determined by comparing $A_0$ and $A_{avg}$[34]:

$$g^{(2)}(\tau = 0) = \frac{A_0}{A_{avg}}. \tag{1}$$

Exciting the QD using LA-phonon-assisted excitation, resonance #1, or resonance #2 leads to single-photon emission with a very low multi-photon contribution as evidenced by $g^{(2)}(\tau = 0) \leq 0.03$. This low multi-photon contribution is a key prerequisite for the implementation of single-photon based applications. The results are summarized in Table 1. Whilst the emission under above-band-gap excitation predominantly exhibits single-photon characteristics with $g^{(2)}(\tau = 0) \leq 0.194 \pm 0.003$, it shows an increased multi-photon contribution compared to the other excitation schemes investigated.

**Photon indistinguishability**

To further assess the quality of the generated photons, we perform two-photon interference measurements using pairs of consecutively emitted photons (see Fig. 1c). The visibility of this interference experiment serves as a direct measure of the photons' indistinguishability[47,48].

± 25 ns. The insets show a zoom-in of the central peaks around $\tau = 0$ for the different excitation schemes. The visibility $V_{TPI}$ is calculated according to Eqn. (2) in a $\Delta t = 3$ ns window (red/green section). The highest visibility of $V_{TPI} = (91.7 \pm 0.2)\%$ is achieved using LA-phonon-assisted excitation.

The results of these two-photon interference measurements are presented in Fig. 4 for the different excitation schemes. The two-photon interference visibility $V_{TPI}$ is calculated as

$$V_{TPI} = 1 - \frac{A_\parallel}{A_\perp}, \tag{2}$$

where $A_\perp$ and $A_\parallel$ are the areas underneath the peak at $\tau = 0$ for orthogonal and parallel input polarization, respectively.

Since the finite multi-photon probabilities $g^{(2)}(\tau = 0)$ affect the visibility, a corrected single-photon indistinguishability $M_s$ can be defined as[42,49]

$$M_s = \frac{V_{TPI} + g^{(2)}(\tau = 0)}{1 - g^{(2)}(\tau = 0)}. \tag{3}$$

The determined values for $V_{TPI}$ and $M_s$ are shown in Fig. 4 and Table 1. Although $V_{TPI}$ for above-band-gap excitation is comparable to previously reported telecom QDs[33,36], it is inherently limited by the higher $g^{(2)}(\tau = 0)$ and the slowly decaying signal present in the correlation (see Fig. 4a).

Amongst all methods, LA-phonon-assisted excitation provides the highest indistinguishability $M_s$.

## Discussion

Whilst all four investigated excitation schemes predominantly exhibit single-photon characteristics, we observe an increased multi-photon contribution under above-band-gap excitation (see Fig. 3). We attribute this to slow refilling of the QD excited state[50]. LA-phonon-assisted excitation significantly outperforms the three other schemes in terms of photon indistinguishability. We attribute this to a reduced state preparation time under LA-phonon-assisted excitation, as is evident from additional lifetime measurements (see Supplementary Fig. 3). We believe that the longer time traces measured for resonance #1, resonance #2 and above-band-gap excitation are caused by slower relaxation channels into the excited state, as we do not think that the type of excitation modifies the excited-state dynamics. Gaining a deeper understanding of the excited-state dynamics and the associated timescales raises interesting questions for future research and is beyond the scope of this study.

We report a QD-based photon source showing two-photon interference visibilities as high as $V_{TPI}$ = 91.7%, thus demonstrating a new benchmark in photon indistinguishability for deterministic photon sources in the telecom C-band[33–36]. Our advancements bring the performance of deterministic quantum emitters in the telecom C-band closer to the thresholds needed for photonic quantum computing technologies[51]. With the reported indistinguishabilities, we close the critical gap between deterministic and probabilistic single-photon sources in the telecom C-band, demonstrating the viability of QD-based devices for photonic quantum technologies in near-term applications.

## Data availability

The authors declare that the data supporting the findings of this study are available within the paper and its Supplementary Information files. The raw data can be shared upon request.

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

## Acknowledgements

We thank J. Kim for identifying the device that has been used in our study. We also thank S. D'Aurelio for helpful discussions and support with the measurement software. We acknowledge support from the Carl Zeiss Foundation, the Center for Integrated Quantum Science and Technology (IQST), the Federal Ministry of Research, Technology and Space (BMFTR, projects SiSiQ: FKZ 13N14920, PhotonQ: FKZ 13N15758, QRN: FKZ 16KIS2207), and the Deutsche Forschungsgemeinschaft (DFG, German Research Foundation, 431314977/GRK2642). Furthermore, we would like to acknowledge support from the state of Bavaria and the Federal Ministry of Research, Technology and Space (BMFTR, projects PhotonQ: FKZ 13N15759, QuNET+ICLink: FKZ 16KIS1975). Tobias Huber-Loyola acknowledges financial support from the BMFTR within the Project Qecs (FKZ: 13N16272).

## Author contributions

N.H., M.B., and S.B. conducted the experiment, analyzed the data, and wrote the main manuscript. J.K., Y.R., G.P., J.M., M.K., T.H-L., A.T.P., and S.H. produced and characterized the sample and prepared parts of the manuscript. All authors discussed the results and reviewed the manuscript. S.H. and S.B. conceived the experiments and jointly supervised the project.

## Funding

## Competing interests

The authors declare no competing interests.
