## [Transparent Peer Review file · Nature Communications]

Deterministic and highly indistinguishable single photons in the telecom C-band

Corresponding Author: Mr Nico Hauser

Editorial Note: Parts of this peer review file have been redacted as indicated to maintain the confidentiality of reviews from other journals.

Version 0:

Reviewer comments:

Reviewer #2

(Remarks to the Author)

see pdf

Reviewer #1 (Remarks to the Author):

I am happy that the authors recognized my comments as constructive feedback and I am satisfied with most of their responses. I am in favor of publication of the current manuscript. However, I want to add 2 points as food for thoughts:

We thank the reviewer for their assessment and their recommendation for publication. We appreciate their further comments on our replies – which demonstrate that our manuscript addresses a very active field of research.

Reviewer: [...]

a) the theoretical 95% is only achieved in a very specific case and would reduce g^2/HOM . The 90% indeed is a theory value, no one ever verified and the temperature of the phonons at the vicinity of the QD is hard to determine. I doubt that the phonon density is so low that 90% can experimentally be achieved in a standard helium cryostat. Without proof I find the current statement in the manuscript not ideal.

b) 90% theoretical value is not “near-unity” from a mathematical standpoint. So even if you reach 90% in the experiment one should not call that near-unity given that still 10% is missing. Especially, since theory forbids unity at finite temperatures, so your experimental value is not limited by measurement uncertainties or artefacts. I would exchange ...ensuring “near-unity” population inversion... with: ...ensuring a population inversion of the quantum system up to 90%.

We would like to thank the reviewer for their reply. We have changed the wording in the manuscript to “[...] allowing for a high excited state occupation of the quantum system.”

[...] I understand the authors point but if you analyze your data like this, you cannot call it $g^2(0)$ value. It is rather the ratio of $P1/P2$. However, this ratio is loss dependent. So, I think it is a very tricky subject. My approach with the normalization ($\tau \rightarrow \infty$) would be a lower bound for the $g^2(0)$ value, whereas yours could be overestimating the number. I am typically more in favor of conservative numbers. In the end the most important thing is that you clearly state how you determine the value you give.

We have addressed this comment by clearly stating in the manuscript how the respective values have been determined.

I am happy that the authors recognized my comments as constructive feedback and I am satisfied with most of their responses. I am in favor of publication of the current manuscript. However, I want to add 2 points as food for thoughts:

[redacted]

My reply: I would disagree with the authors based on 2 points:

a) the theoretical 95% is only achieved in a very specific case and would reduce g^2/HOM . The 90% indeed is a theory value, no one ever verified and the temperature of the phonons at the vicinity of the QD is hard to determine. I doubt that the phonon density is so low that 90% can experimentally be achieved in a standard helium cryostat. Without proof I find the current statement in the manuscript not ideal.

b) 90% theoretical value is not “near-unity” from a mathematical standpoint. So even if you reach 90% in the experiment one should not call that near-unity given that still 10% is missing. Especially, since theory forbids unity at finite temperatures, so your experimental value is not limited by measurement uncertainties or artefacts.

I would exchange ...ensuring “near-unity” population inversion... with: ...ensuring a population inversion of the quantum system up to 90%.

[redacted]

My reply: I understand the authors point but if you analyze your data like this, you cannot call it $g_2(0)$ value. It is rather the ratio of P_1/P_2 . However, this ratio is loss dependent. So, I think it is a very tricky subject. My approach with the normalization ($\tau \rightarrow \infty$) would be a lower bound for the $g_2(0)$ value, whereas yours could be overestimating the number. I am typically more in favor of conservative numbers. In the end the most important thing is that you clearly state how you determine the value you give.